# Identifying Drivers of COVID-19 Vaccine Uptake among Residents of Yopougon Est, Abidjan, Côte d’Ivoire

**DOI:** 10.3390/vaccines10122101

**Published:** 2022-12-08

**Authors:** Brian Pedersen, Katherine Thanel, Albert Yao Kouakou, Jariseta Rambeloson Zo, Mariame Louise Ouattara, Dorgeles Gbeke, Gretchen Thompson, Sohail Agha

**Affiliations:** 1Department of Social and Behavior Change, FHI 360, Washington, DC 20009, USA; 2Independent Research Consultant, Abidjan 00225, Côte d’Ivoire; 3Department of Social Sciences and Humanities, University of Jean Lorougnon Guédé of Daloa, Sassandra-Marahoué District, Daloa 150, Côte d’Ivoire; 4Department of Behavioral, Epidemiological and Clinical Sciences, FHI 360, Durham, NC 27701, USA; 5Behavior Design Lab, Stanford University, Stanford, CA 94305, USA

**Keywords:** COVID-19 vaccination, Fogg Behavioral Model, motivation, ability

## Abstract

This study applied the Fogg Behavioral Model (FBM) to identify and prioritize factors influencing COVID-19 vaccination among residents of Yopougon Est, Abidjan, Côte d’Ivoire. A total of 568 respondents were recruited from among individuals entering eleven participant recruitment and data collection sites located near high pedestrian trafficked areas. Among all respondents, 52% reported being vaccinated versus 48% who reported not being vaccinated. Of those who reported being vaccinated, 42% reported received a single dose, 54% a double dose, and 4% three or more doses. A categorical regression analysis suggested that potential predictors of COVID-19 vaccination included acceptance and rejection factors, which are both aligned with motivation in the FBM and socio-demographic characteristics, proximity to services, and religion. Our findings suggest that demand creation activities should target individuals with less formal education, those who are not formally employed, non-Catholic Christians, and individuals who do not identify as Akan. Results also suggest the need to design programmatic messages and activities that focus on generating family and community support for COVID-19 vaccination.

## 1. Introduction

Since the start of the global COVID-19 pandemic in March 2020 and up to October 2022, 55 African Union Member States have together reported over 12,000,000 confirmed cases of COVID-19 [1]. Early reporting suggested that while COVID-19 vaccines had been made available in many countries in Africa, vaccination coverage had been inconsistent [2]. To boost vaccination coverage in Côte d’Ivoire, additional COVID-19 vaccine supplies were made available in late 2021. Despite early success in increasing daily vaccination rates, by 18 September 2022, only 33.43% of Ivorians were fully vaccinated [3]. High throughput fixed and mobile vaccination sites designed to reduce financial and convenience barriers to access have been used to ensure the efficient and safe delivery of COVID-19 vaccines to populations living in Abidjan.

The success of these vaccination sites relies on strategies to capacitate and motivate clients to access them. The COVID-19: Social Marketing and Behavioral Science Tools and Approaches for Optimizing Throughput at Mass Vaccination Sites in Sub-Saharan Africa project was designed to support service delivery partners in the Yopougon Est commune of Abidjan the commercial capital of Côte d’Ivoire. Yopougon Est is the largest and most densely populated commune in Côte d’Ivoire with over 1.5 million inhabitants made up of nearly every ethnic group within Côte d’Ivoire [4,5]. The commune includes both residential and industrial zones and is considered an important political and economic center within the country [5].

Our project aimed to increase uptake of COVID-19 vaccines in Yopougon Est using behavioral science approaches to design and implement social and behavior change interventions aligned with client attitudes, preferences, and needs. The project applied a standard process that begins with audience insight generation activities to inform the design of demand creation campaigns. These demand creation campaigns used radio spots, posters and pamphlets, social media postings, engagement with local associations, and interpersonal communication to engage target audiences and address their ability and motivation needs to seek out COVID-19 vaccination. Campaign activities were implemented in partnership with service delivery partners to ensure coordination with vaccination services.

A key best practice when designing social and behavior change interventions is the application of behavioral theories and models to improve the effectiveness of interventions by linking them to critical factors influencing the adoption of the promoted behavior [6]. The Fogg Behavioral Model (FBM) is a simple behavioral model that posits that adoption of a behavior occurs when a person is sufficiently motivated, has the ability, and is prompted to perform the behavior [7]. The FBM has been used to inform COVID-19 prevention messaging in Saudi Arabia [8] and to understand drivers of COVID-19 vaccine uptake among healthcare workers in Nigeria [9]. We applied the FBM to inform formative research and subsequent design of a campaign to increase uptake of COVID-19 vaccination in Yopougon Est, Abidjan, and we present the results of our analysis in this paper.

## 2. Materials and Methods

### 2.1. Study Design and Data Collection

Our study used a cross-sectional design. Kasi Insight, a market research firm, recruited a convenience sample from among individuals entering eleven participant recruitment and data collection sites co-located within Internet cafés near high pedestrian trafficked areas such as shopping malls and transportation hubs. Participants were aged 18 to 54 and resided in Yopougon East, Abidjan. Trained recruiters were positioned at the entry of these Internet cafés in morning and afternoon hours from Monday to Saturday. These recruiters approached potential respondents entering the Internet cafés who they believed fit the defined inclusion criteria. Recruiters were allocated daily recruitment quotas based on neighborhood, age, and gender to match the demographic breakdown of the commune. The recruiter would ask the individual for their age and where they live to confirm they fell within an unfilled quota. If the individual agreed to participate, they were invited to approach a computer terminal where they would complete the survey. No paper survey was offered. A sample size of 500 was informed by the target population size, as well as the nature of the analysis to be conducted, and was based on an expected respondent vaccination rate of 20.5%, which was the estimated COVID-19 vaccination rate in Côte d’Ivoire as of 26 April 2022 [10]. The survey was conducted in September 2022.

### 2.2. Questionnaire

The structured questionnaire consisted of 41 multiple-choice questions or scale items, aligned with the FBM, and operationalized variables for motivation and ability. Scale items to measure ability factors were adapted from a 2021 study to understand drivers of COVID-19 vaccination uptake among health care providers in Nigeria [9]. Scale items to measure motivation factors were developed by the project team using results from qualitative research completed in February 2022. These scales were developed using cognitive interviews to assess comprehension, confidence in response, level of discomfort and social desirability, and to determine whether these scale items, as well as those adapted for ability, accurately measured the factors of interest for people living in Yopougon Est.

### 2.3. Outcomes and Variables

The expected outcome of this study was data to inform demand creation messages and activities aligned with the FBM elements of motivation and ability most likely to influence COVID-19 vaccine uptake. Specifically, this study aimed to determine which FBM elements of motivation and ability and sociodemographic characteristics are correlated with COVID-19 vaccination intention and completion of recommended COVID-19 vaccination regimens.

Vaccination status was the dependent variable and was measured using the following question: “What is your vaccination status?” Additional questions were asked of those who answered “vaccinated” to specify where they were vaccinated, which vaccine(s) they had received, and how many doses they had received.

Socio-demographic variables included age, gender, level of education, employment status, type of employment (if employed), place of residence within Yopougon Est, marital status, religious affiliation, income, and ethnic group identity. Motivation and ability were first measured using two general questions: motivation (“to what extent do you think it is important for your health to get a COVID-19 vaccine”) was measured on a 5-point Likert scale ranging from 1 = not at all important to 5 = very important and ability (“to what extent do you think it is easy or difficult to get a COVID-19 vaccine”) was measured on a 5-point Likert scale ranging from 1 = very difficult to 5 = very easy.

In addition, 12 scale items for six dichotomous motivation factors (acceptance/rejection, hope/fear, and pleasure/pain) and 5 scale items for five ability factors (time, cost, physical effort, mental effort, and routine) were measured on a 5-point Likert scale ranging from 1 = strongly disagree to 5 = strongly agree. Other questions measuring exposure to COVID-19 messages (promoting preventive actions and vaccination), media preferences, trusted sources of COVID-19 information, and awareness of vaccination sites were used to collect information that could be used to inform the selection of campaign communication channels.

### 2.4. Analysis

Among self-reported vaccinated respondents (n = 601), only those who reported a vaccination schedule aligned with current national COVID-19 vaccination protocols were included (n = 568). For example, participants who reported being vaccinated with three or more doses of the Johnson & Johnson vaccine only, i.e., not combined with AstraZeneca and/or BioNTechPfizer, were excluded from our analyses. Responses from the remaining 568 participants were retained for descriptive statistical analyses, which were supplemented by a Pearson chi-square test to determine whether the frequencies of the variables disaggregated by vaccination status differed significantly (a threshold of *p*-value < 0.001 was retained). Next, the internal reliability (of Likert scale responses) of the 12 scale items related to motivation factors and 5 related to ability factors were assessed separately. Reliability of each group was determined by a Cronbach’s Alpha score of >0.7 [11]. An initial classification was then performed using the Naive Bayes method to identify a preliminary list of predictors (20 in number). We refined this list by a second reduction method, using the “optimal scaling” option of the categorical regression (CATREG). The weight and importance of the potential predictors was determined by a significance coefficient of <0.001, the magnitude of variable coefficients, as well as the absence of multicollinearity between the different selected predictors determined by coefficients of tolerance before and after transformations close to 1. All analyses and statistical tests were carried out using SPSS version 26 [12]. This study was determined exempt by FHI 360′s Protection of Human Subjects Committee and was approved by the Ivorian Ministry of Health’s National Research Ethics Committee.

## 3. Results

### 3.1. COVID-19 Vaccination Status

The questionnaire was completed by 601 respondents although the results below represent the 568 respondents remaining after data cleaning. Figure 1 shows that 52% of respondents reported being vaccinated versus 48% who reported not being vaccinated. The AstraZeneca vaccine is the most widely reported (48%), followed by BioNTech Pfizer (42%), and Johnson & Johnson (13%). Of those who reported being vaccinated, 42% reported received a single dose, 54% a double dose, and 4% three or more doses.

### 3.2. COVID-19 Vaccination Status by Socio-Demographic Characteristics

Table 1 presents vaccination status by sample socio-demographic characteristics. A similar proportion of male respondents (53.2%) as female respondents (54.5%) reported being vaccinated. We observe that age does not seem to be significantly associated with vaccination status among our sample.

Individuals who have obtained some form of secondary education were over one and a half times more likely (OR = 1.779, *p* < 0.05) to be vaccinated (66.1%) than unvaccinated (33.9%). Among individuals with technical or professional qualifications, a greater proportion were unvaccinated (56%) than vaccinated (44%), although this difference was not statistically significant limiting our observation to the sample only. Those who were formally employed full-time were over twice as likely (OR = 2.170, *p* < 0.05) to be vaccinated (67.2%) than unvaccinated (32.8%). Conversely, among those who identified as self-employed workers were less likely to be vaccinated (OR = 0.404, *p* < 0.05), with a much lower proportion of vaccinated (35.3%) compared to unvaccinated (64.7%) respondents.

Similarly, proximity to services was significantly (*p* < 0.001) associated with vaccination status. Individuals living in areas served through mobile vaccination services (Sideci, Niangon, Toits Rouge, Maroc, and Sicogi) were four times more likely to be vaccinated than those living in other locations. In contrast, few respondents living in the immediate vicinity of the mass COVID-19 vaccination center were vaccinated (OR = 0.143, *p* < 0.001).

Furthermore, respondents who identify as Catholic were nearly twice as likely to be vaccinated (65%) than unvaccinated (34.4%) (OR = 1.954, *p* < 0.001), while those who identify as adherents of traditional African religions were less likely to be vaccinated (33.3%) than unvaccinated (66.7%) (OR = 0.417, *p* < 0.05). Finally, only those who identify as Akan were significantly more likely to report being vaccinated (61.1%) than unvaccinated (38.9%) (OR = 1.525, *p* < 0.05).

### 3.3. COVID-19 Vaccination Status by General Measures of Motivation and Ability

Belief in the importance of vaccination (motivation) and positive perceptions about the ease of vaccination (ability) were both significantly associated (*p* < 0.001) with vaccination status. As shown in Table 2, vaccinated respondents had a higher mean score for motivation (4.26) and ability to vaccinate (4.02) than unvaccinated respondents.

### 3.4. COVID-19 Vaccination Status by Motivation and Ability Measures

Ten of the twelve scale items used to measure motivation factors were significantly associated with vaccination status. Table 3 shows higher mean scores for acceptance, hope, pleasure, and fear among vaccinated respondents and higher mean scores for rejection and pain among unvaccinated respondents. In addition, the overall high Cronbach’s Alpha score of 0.742 demonstrates consistency and acceptable reliability of the group of 12 scale items.

Four of the five scaled items used to measure ability factors were significantly associated with vaccination status. As shown in Table 4, higher mean scores are observed for time, cost, physical effort, and mental effort among unvaccinated respondents and a higher mean score for routine among vaccinated respondents. The overall high Cronbach’s Alpha score of 0.702 demonstrates consistency and acceptable reliability of the group of five scaled items.

### 3.5. Categorical Regression-Based Classification to Identify Potential Predictors of COVID-19 Vaccination Status

To determine which, if any, socio-demographic and FBM motivation and ability factors remain significantly associated with COVID-19 vaccination after controlling for other variables, we conducted a categorical regression analysis. Based on this analysis, the potential predictors of COVID-19 vaccination, as presented in Table 5, included acceptance and rejection factors, which are both aligned with motivation in the FBM, and socio-demographic characteristics, proximity to services, and religion. The largest importance corresponded to general motivation (36.0%). General motivation, proximity to services and acceptance (my family wants me to get vaccinated against COVID-19) account for 95% of the importance for this combination of potential predictors.

## 4. Discussion

To our knowledge, our project is the first to apply the FBM to understand and design interventions to address factors influencing COVID-19 vaccination in Côte d’Ivoire. In our study, the regression analysis indicates that those with higher perceived family support for vaccination were significantly more likely to be vaccinated than unvaccinated. Additionally, bivariate analysis confirmed that level of education, type of employment, proximity to services, marital status, religion, and ethnic group are significantly associated with COVID-19 vaccination status.

Contrary to the findings of other studies [13,14,15,16], our study did not find a significant association between gender, age, and COVID-19 vaccination status. One study conducted in Ethiopia [17] similarly concluded that gender was not significantly associated with willingness to receive COVID-19 vaccination, which may suggest that the way gender influences COVID-19 vaccination is highly context specific. However, as suggested elsewhere [13], our study did find that level of education was significantly associated with COVID-19 vaccination status. Respondents with secondary level education or higher were more likely to be vaccinated than those with less than secondary education and those with only technical or professional qualifications although these findings were not significant. Other studies have found that lower education levels are significantly associated with greater COVID-19 vaccine hesitancy [18,19].

There are contradicting conclusions on whether employment status may predict COVID-19 vaccination status [20,21]. However, our study found that respondents who reported full-time, formal employment were significantly more likely to be vaccinated than respondents who reported being self-employed. As demonstrated by Fishman in a randomized, controlled survey experiment [22], these findings may indicate the impact of mandates introduced by employers for those returning to workplaces after the introduction of COVID-19 vaccines in Abidjan. These findings are bolstered by anecdotal evidence in Yopougon Est, where factories providing full-time employment have routinely mandated vaccination. It is therefore not surprising that self-employed individuals are significantly less likely to be vaccinated than vaccinated.

While Goralnick [23] argues that mass COVID-19 vaccination sites are an important strategy to increase coverage, our study found that respondents reporting proximity to the fixed high throughput COVID-19 vaccination site in Yopougon Est were significantly less likely to be vaccinated. This may indicate that the effects of popular protests experienced at the installation of that site [24], first as a COVID-19 testing facility, may continue to influence COVID-19 hesitancy among residents living in its immediate vicinity. This finding reinforces the need for adequate community consultation and engagement prior to the installation of COVID-19 services, something Gilmore [25] has described as a “fundamental component” of any successful epidemic control effort.

Importantly, our study found that respondents living in areas served by mobile COVID-19 vaccination outreach teams were over four times as likely to be vaccinated than those living in other areas which aligns with the ability proposition of the FBM and highlights the importance of maintaining mobile outreach service approaches to ensure ease of access.

As suggested elsewhere [17,26], our study found that married respondents were significantly more likely to be vaccinated than not. Religion was only found to be significantly associated with vaccination status among respondents who identified as Catholic or adherents of traditional African religions. In those cases, Catholic respondents were more likely to be vaccinated while adherents of traditional African religious were less likely. Respondents identifying as Methodist, Protestant, and Muslim were less likely to be vaccinated and those identifying as Evangelical were more likely to be vaccinated, although these findings were not significant. To some extent, these findings align with a cross-national comparison by Trepanowski [27] which suggested that among religions, only Christianity was a significant predictor of vaccination status.

Similar to findings from Nigeria and Uganda [28,29], our study found that identification with certain ethnic groups was associated with COVID-19 vaccination status. Specifically, we found that those who identified as Akan were significantly more likely to be vaccinated while those who identified with other ethnic groups (Gour, Krou, and Mandé) were less likely to be vaccinated, although these findings were not significant.

As expected, the results of our bivariate analysis support the central hypothesis of the FBM: respondents who reported higher motivation (“To what extent do you think it is important for your health to get a COVID-19 vaccine?”) and higher ability (“To what extent do you think it is easy or difficult to get a COVID-19 vaccine?”) to vaccinate against COVID-19 were more likely to be vaccinated than unvaccinated. This general conclusion is consistent with other studies that have applied the FBM to understand drivers of other behaviors, including COVID-19 vaccination [9].

To our knowledge, this was the first study in Africa to use multiple scale items to measure FBM motivation factors. Prior studies have relied on one question to measure motivation (“How important is it for you, personally, to …?”) [30], but we have argued that those results do not provide sufficient nuance to inform the design of activities to address low motivation. Instead, as described above, this project developed and applied scale items to measure each of the six FBM motivation factors. This allowed us to indicate which of the motivation factors best explained underlying contributors to motivation without requiring additional research.

The result is apparent in the categorical regression analysis which found that among the six FBM motivation factors, acceptance and rejection, indicators of social support, were significant predictors of COVID-19 vaccination. The role social support plays in behavior change has been described and theorized for several decades [31], and our findings align with more recent studies that have confirmed its relevance to COVID-19 vaccination [32,33,34]. Specifically, perceptions of the way a respondent’s family would react, either positively or negatively, to their decision to receive a COVID-19 vaccination is a key predictor of uptake.

This study was conducted as an experiment requested by the BMGF to explore whether private-sector market research methods could produce sufficient insight for ongoing program design. These methods and associated cost savings were prioritized over a design with a random sampling strategy. We recognize the bias inherent in the sampling approach and the resulting inability to generalize findings.

A convenience sample focused on individuals in or near the eleven data collection sites co-located within internet cafés. It is likely respondents frequenting Internet cafés represent a unique segment of the population. We hypothesize they are more literate and likely to have some disposable income compared with the general population, yet unlikely to be wealthy if they rely on cafés to access the Internet rather than mobile phones or home service. Respondents are likely different in this and other unmeasurable ways from the general population.

Strict confidentiality procedures assured responses were entered directly into a computer, and not viewed by anyone other than the respondent. Despite these efforts, the sensitivities around COVID-19 vaccination in Yopougon Est may have skewed our sample towards vaccinated individuals, as recruiters reported unvaccinated individuals declining the survey once they learned of the topic out of fear that they would be identified and later contacted for vaccination.

We kept the survey as short as possible to prioritize completion rates and data quality. The result is that we were not able to measure other topics, such as trust in institutions, which have shown to be associated with vaccine hesitancy in other studies.

## 5. Conclusions

Our study demonstrates that the FBM is an effective model to understand and design interventions to address motivation and ability factors influencing COVID-19 vaccination among residents of Yopougon Est, Abidjan, Côte d’Ivoire. Analysis of the FBM motivational factors (acceptance/rejection, hope/fear, pleasure/pain) and ability factors (time, cost, physical effort, mental effort, routine) coupled with socio-demographic characteristics address the evidence requirements to establish evidence-based demand creation objectives and develop targeted interventions.

Our findings on socio-demographic characteristics associated with vaccination status suggest that demand creation activities should engage individuals with less formal education and those who do not have full-time, formal employment. This may require approaches implemented in community settings (rather than workplaces) using culturally appropriate, low-literacy or visual materials and tools. Additional effort may also be required to overcome lingering COVID-19 vaccine hesitancy among individuals living in the immediate vicinity of the fixed high throughput COVID-19 vaccination site. Finally, activities specifically tailored to engage non-Catholic Christians and individuals who do not identify as Akan may be required to overcome lower COVID-19 vaccination tendencies in these groups. Results demonstrating the predictive power of acceptance and rejection have suggest the need to design programmatic messages and activities that focus on generating family and community support for COVID-19 vaccination.

## Figures and Tables

**Figure 1 vaccines-10-02101-f001:**
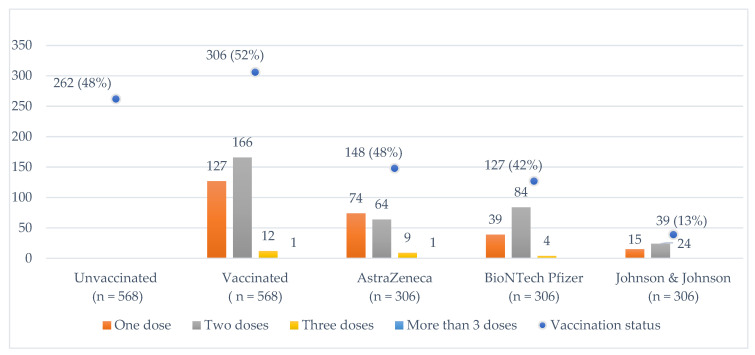
Reported vaccination status, type of vaccine *, and number of doses received. * Type of vaccine received is not cumulative since multiple responses are possible due to mixing (e.g., an individual receiving two doses may have been administered AstraZeneca and Johnson & Johnson).

**Table 1 vaccines-10-02101-t001:** COVID-19 vaccination status by socio-demographic characteristics.

Characteristic	Total	Vaccinated	Unvaccinated	OR	Sig
	n	%	%		
Gender					
Male	293	53.2	46.8		
Female	275	54.5	45.5	0.949	ns
Age					
18–24 ^α^	150	56	44	1.124	ns
25–29 ^α^	106	55.7	44.3	1.093	ns
30–34 ^α^	84	48.8	51.2	0.788	ns
35–39 ^α^	77	57.1	42.9	1.165	ns
40–44 ^α^	70	48.6	51.4	0.785	ns
45–59 ^α^	81	54.3	45.7	1.021	ns
Level of education					
No education, or less than secondary ^ß^	126	53.2	46.8	0.965	ns
Some secondary ^ß^	62	66.1	33.9	1.779	*
Secondary diploma or higher ^ß^	244	56.1	43.9	1.178	ns
Technical or professional qualification ^ß^	125	44	56	0.600	ns
Type of employment					
Formally employed, full-time ^ɣ^	67	67.2	32.8	2.170	*
Formally employed, part-time ^ɣ^	27	55.6	44.4	1.101	ns
Self-employed ^ɣ^	51	35.3	64.7	0.404	*
Business owner ^ɣ^	34	50	50	0.856	ns
No formal employment ^ɣ,a^	100	54	46	1.038	ns
Proximity to services					
Areas served through mobile services ^ε,b^	245	72.7	27.3	4.047	**
Areas close to vaccination center ^ε,c^	191	24.1	75.9	0.143	**
Other places ^ε,d^	132	62.1	37.9	1.552	*
Marital status					
Single without girlfriend/boyfriend ^Ω^	106	51.9	48.1	0.907	ns
Single with girlfriend/boyfriend ^Ω^	188	54.3	45.7	0.720	ns
Cohabitating ^Ω^	162	48.8	51.2	0.749	ns
Married ^Ω^	101	63.4	36.6	1.612	*
Religion					
Catholic ^∆^	157	65.6	34.4	1.954	**
Methodist ^∆^	67	47.8	52.2	0.757	ns
Evangelical ^∆^	118	54.2	45.8	1.018	ns
Protestant ^∆^	44	45.5	54.5	0.693	ns
Muslim ^∆^	116	50.9	49.1	0.859	ns
Traditional African ^∆^	18	33.3	66.7	0.417	*
Ethnic group					
Akan ^Φ^	190	61.1	38.9	1.525	*
Gour ^Φ^	101	50.5	49.5	0.832	ns
Krou ^Φ^	118	51.7	48.3	0.878	ns
Mandé ^Φ^	148	50	50	0.794	ns

** *p* < 0.001, * *p* < 0.05, ns = not significant. ^α^ other age group is reference, ^ß^ other education level is reference, ^ɣ^ other type of employment is reference, ^ε^ other area is reference, ^Ω^ other marital status is reference, ^∆^ other religion is reference, ^Φ^ other ethnic group is reference. ^a^ student, stay at home mother or father, unemployed, retired; ^b^ Sideci, Niangon, Toit Rouge, Maroc, Sicogi; ^c^ Segbe, Camp Militaire, Sapeur Pompiers; ^d^ Annaneraie, Millionaire, Autres.

**Table 2 vaccines-10-02101-t002:** COVID-19 vaccination status by general motivation and ability measures.

General Measure	Vaccinated	Unvaccinated	Sig
	Mean	Mean	
	n = 306	n = 262	
To what extent do you think it is important for your health to get a COVID-19 vaccine?	4.26	3.18	**
To what extent do you think it is easy to get a COVID-19 vaccine?	4.02	3.15	**

** *p* < 0.001.

**Table 3 vaccines-10-02101-t003:** COVID-19 vaccination status by motivation measures.

Motivation Measures	Vaccinated	Unvaccinated	Sig
	Mean	Mean	
	n = 306	n = 262	
Acceptance			
My family wants me to get vaccinated against COVID-19.	3.80	2.87	**
Getting vaccinated against COVID-19 would make me feel more accepted by the people around me.	3.44	2.84	**
Rejection			
My family would be angry with me if I got vaccinated against COVID-19.	2.64	3.09	**
Most of the people I know would think poorly of me if I were to get a COVID-19 vaccine.	2.82	2.94	ns
Hope			
Getting vaccinated against COVID-19 would protect me from getting sick.	3.78	3.27	**
Getting vaccinated against COVID-19 would allow me to keep my job.	3.45	2.79	**
Fear			
I worry about getting ill from COVID-19.	4.11	3.50	**
I worry about COVID-19 infecting someone in my family.	4.23	3.54	**
Pleasure			
I would feel more at ease everyday if I were vaccinated against COVID-19.	3.91	3.16	**
It would make me feel good knowing that I am protecting my family by getting vaccinated against COVID-19.	4.12	3.13	**
Pain			
I worry that the COVID-19 vaccine will make me sick.	3.68	3.90	*
I worry that the needlestick for COVID-19 vaccine will be painful.	3.38	3.53	ns
Cronbach’s Alpha	0.742		

** *p* < 0.001, * *p* < 0.05, ns = not significant.

**Table 4 vaccines-10-02101-t004:** COVID-19 vaccination status by ability measures.

Ability Measures	Vaccinated	Unvaccinated	Sig
	Mean	Mean	
	n = 306	n = 262	
Time: My family and work responsibilities make it difficult for me to find time to get a COVID-19 vaccine.	2.73	2.90	ns
Cost: Costs associated with getting the COVID-19 vaccine are keeping me from getting it.	2.63	2.96	**
Physical effort: The distance I must travel to get the COVID-19 vaccine keeps me from getting vaccinated.	2.68	2.91	*
Mental effort: The decision to get the COVID-19 vaccine is difficult.	3.14	3.46	**
Routine: The vaccine is available in places that I routinely visit.	3.64	3.36	**
Cronbach’s Alpha	0.702		

** *p* < 0.001, * *p* < 0.05, ns = not significant.

**Table 5 vaccines-10-02101-t005:** Potential predictors of COVID-19 vaccination and predicted descriptive values.

Potential Predictors	Vaccinated	Unvaccinated				Coefficients	
	N = 302	N = 258	F	Sig	Importance	Beta	Sig
	Mean	Mean					
Acceptance: My family wants me to get vaccinated against COVID-19.	3.80	2.87		**	0.277	−0.245	**
Rejection: Most of the people I know would think poorly of me if I were to get a COVID-19 vaccine.	2.81	2.93		ns	0.009	0.050	*
General motivation: To what extent do you think it is important for your health to get a COVID-19 vaccine.	3.27	2.36		**	0.360	−0.256	**
	**%**	**%**	**OR**	**Sig**			
Proximity to services					0.313	0.358	**
Areas served through mobile services ^ε,b^	72.7	27.3	4.092	**			
Areas close to vaccination center ^ε,c^	23.0	77.0	0.134	**			
Other places ^ε,d^	62.1	37.9	1.551	*			
Religion					0.041	0.122	**
Catholic ^∆^	65.6	34.4	1.955	**			
Methodist ^∆^	47.8	52.2	0.755	ns			
Evangelical ^∆^	54.2	45.8	1.016	ns			
Protestant ^∆^	45.5	54.5	0.691	ns			
Muslim ^∆^	45.0	50.0	0.680	ns			
Traditional African ^∆^	33.3	66.7	0.416	*			

(R^2^ = 0.419, F = 39.570, Sig **). ** *p* < 0.001, * *p* < 0.05, ns = not significant. ^ε^ other area is reference, ^∆^ other religion is reference, ^b^ Sideci, Niangon, Toit Rouge, Maroc, Sicogi; ^c^ Segbe, Camp Militaire, Sapeur Pompiers; ^d^ Annaneraie, Millionair.

## Data Availability

De-identified aggregate data is available upon request.

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
