# Peer review of "Identifying Drivers of COVID-19 Vaccine Uptake among Residents of Yopougon Est, Abidjan, Côte d’Ivoire"

_vaccines, 2022, doi:10.3390/vaccines10122101_

Round 1
Reviewer 1 Report
The authors have presented a very detailed and well-written paper on using a behavioural model for acceptance of COVID-19 vaccines. I recommed this paper for publsihing after checking for minor English errors in the paper.
Author Response
Point 1: Checking for minor English errors in the paper
Response 1: A thorough copyedit of the manuscript has been completed and grammatical/spelling problems have been resolved.
Reviewer 2 Report
Presented manuscript describes intresting behavioral study concerning preferences for COVID-19 vaccination. I see this study as high quality, well designed research aiming to answer for important society question, i.e. what can be done to increse COVID-19 vaccinaton rate. Therefore I recommend this manuscript to be published in this Journal. To improve this manuscript I recommend to summarize the most important factors influencing COVID-19 vaccination in graphical way.
Author Response
Point 1: To improve this manuscript I recommend to summarize the most important factors influencing COVID-19 vaccination in graphical way.
Response 1: Unfortunately, we are unable to propose a graphical representation of the most important factors identified in our study. However, we feel that tables 3, 4, and 5 provide the reader easy-to-understand summaries of the most important findings from the study.
Reviewer 3 Report
the manuscript is novel and well written, and the results are presented clearly. so recommended for publication
Author Response
No points presented.
Reviewer 4 Report
1. These authors used the Fogg Behavioral model to identify and prioritize factors which might influence COVID vaccination in Yopougon Est. The study included 568 respondents who were recruited at 11 participant recruitment and data collection sites located in high pedestrian traffic areas in the city. 52% of the participants had been vaccinated. They were able to identify the characteristics of participants who have been vaccinated. Based on analysis of participants who had not been vaccinated, they concluded that vaccine campaigns should target individuals with less formal education, those were not formally employed, and non-Catholic Christians.
2. These authors accomplished their study goals. The analysis seems correct. They generated definite conclusions which could inform a policy. It would be helpful if they would add a few more details to the study design. On what days of the week were participants recruited? What was the time period during which participants were recruited? Was there a particular quota for a given day so that there was no disproportionate influence of one collection period? Approximately what percentage of the possible participants approached regarding this survey agreed to participate? In the study design section, the authors state that if an individual agreed to participate he or she was invited to a project computer terminal with the survey. Was this survey on a computer or on paper or on both? What if the individual was not used to using a computer? Did the recruiter help the participant if they were unable to use a computer easily? Does this approach potentially influence the results?
3. Many readers will not know much about this region of the world. It might be helpful if the authors added a few additional details regarding this city.
Author Response
Point 1: It would be helpful if they would add a few more details to the study design.
Response 1: Thank you for this suggestion. Additional information about participant recruitment, daily quotas, and how surveys were completed has been added to lines 75-85. Unfortunately, the number of individuals who were invited but refused to participate in the study was not collected systematically so we are unable to provide a refusal rate.
Point 2: Many readers will not know much about this region of the world. It might be helpful if the authors added a few additional details regarding this city.
Response 2: Thank you for this observation. Additional details about the commune where the study was conducted have been added to lines 43-47.